# 1,2,3,4,6-O-Pentagalloylglucose Protects against Acute Lung Injury by Activating the AMPK/PI3K/Akt/Nrf2 Pathway

**DOI:** 10.3390/ijms232214423

**Published:** 2022-11-20

**Authors:** Qi Zhang, Sai Cheng, Zhiming Xin, Haohua Deng, Ying Wang, Qiang Li, Gangwei Wu, Wei Chen

**Affiliations:** 1Department of Natural Medicine, School of Pharmacy, Fujian Medical University, Fuzhou 350122, China; 2Fujian Research Center of Drug’s Non-Clinical Safety Evaluation, Fujian Medical University, Fuzhou 350122, China; 3Department of Pharmaceutical Analysis, School of Pharmacy, Fujian Medical University, Fuzhou 350122, China; 4School of Chinese Materia Medica, Beijing University of Chinese Medicine, Beijing 100102, China; 5Department of Pharmacy, Fujian Provincial Hospital, Fuzhou 350122, China

**Keywords:** 1,2,3,4,6-O-pentagalloylglucose, acute lung injury, AMPK/PI3K/Akt/Nrf2 signaling pathway

## Abstract

An acute lung injury (ALI) is a serious lung disease with a high mortality rate, warranting the development of novel therapies. Previously, we reported that 1,2,3,4,6-O-pentagalloylglucose (PGG) could afford protection against ALI, however, the PGG-mediated protective effects remain elusive. Herein, PGG (60 and 30 mg/kg) markedly inhibited the lung wet/drug weight ratio and attenuated histological changes in the lungs (*p* < 0.05). A pretreatment with PGG (60 and 30 mg/kg) reduced the number of total leukocytes and the production of pro-inflammatory cytokines IL-6 and IL-1β in bronchoalveolar lavage fluid (*p* < 0.05). In addition, PGG (60 and 30 mg/kg) also attenuated oxidative stress by reducing the formation of formation and the depletion of superoxide dismutase to treat an ALI (*p* < 0.05). To further explore the PGG-induced mechanism against an ALI, we screened the PGG pathway using immunohistochemical analysis, immunofluorescence assays, and Western blotting (WB). WB revealed that the expression levels of adenosine monophosphate-activated protein kinase phosphorylation (p-AMPK), phosphoinositide 3-kinase (PI3K), protein kinase B phosphorylation (P-Akt), and nuclear factor erythroid 2-related factor (Nrf2) were significantly higher in the PGG group (60 and 30 mg/kg) than in the lipopolysaccharide group (*p* < 0.05); these findings were confirmed by the immunohistochemical and immunofluorescence results. Accordingly, PGG could be effective against an ALI by inhibiting inflammation and oxidative stress via AMPK/PI3K/Akt/Nrf2 signaling, allowing for the potential development of this as a natural drug against an ALI.

## 1. Introduction

An acute lung injury (ALI) is a serious acute respiratory disease caused by various internal and external pathogenic factors, in addition to cardiogenic factors (including severe acute respiratory syndrome [SARS] and SARS-coronavirus 2 [SARS-CoV-2]). An ALI is characterized by progressive hypoxemia and respiratory distress, which can easily transform into acute respiratory distress syndrome (ARDS). An ALI and ARDS are persistent pathophysiological processes, often described as ALI/ARDS in clinical settings. However, despite the rapid progress in trauma resuscitation technology, the mortality rate of ALI/ARDS remains at 40–46% [1,2]. The COVID-19 pandemic has become a serious public health challenge worldwide. ARDS and refractory hypoxemia are the leading causes of death associated with COVID-19, owing to severe pulmonary inflammatory lesions, namely ALI/ARDS [3,4]. Thus, blocking or reversing an ALI could be an effective strategy for treating ALI/ARDS. 

Several currently available drugs are used to treat an ALI [5,6,7]. First, glucocorticoids are the main drugs used to treat an ALI by regulating the inflammatory response; however, an improper dosing and medication duration can increase the risk of mortality. Second, nitric oxide drugs can dilate pulmonary blood vessels and improve the patient’s oxygenation, but they fail to reduce the death rate. In addition, a supplementation with surfactants could shorten the duration of ventilation but not reduce the mortality. Emerging cytokine inhibitors and stem cell therapies are yet to be established as routine therapies. Accordingly, novel, more suitable drugs are urgently required to treat an ALI.

An ALI has been associated with several pathogenic factors, exhibiting complex pathogenesis. The treatment of an ALI can be challenging and has become a hot topic in critical care medicine. Moreover, the relationship between oxidative stress and an ALI has garnered considerable attention. In the human body, a stimulation by various harmful substances and the induction of oxidative stress can cause an imbalance in oxidative and antioxidant systems. Reactive oxygen species (ROS) are generated in large quantities and accumulate in the body, which can hinder their clearance, subsequently resulting in tissue damage. In ALI rats, the failure to remove accumulated ROS promptly is a key factor leading to pulmonary endothelial cell apoptosis [8]. To prevent oxidative stress, cells must activate and engage antioxidant defense systems to resist ROS, inducing endogenous antioxidant substances such as superoxide dismutase (SOD) and malondialdehyde (MDA) [9].

1,2,3,4,6-O-pentagalloylglucose (PGG) is a potent antioxidant that belongs to the tannin group, known to exist in several traditional Chinese medicines, such as *Paeonia lactiflora* Pall. and *Paeonia suffruticosa* Andr. (Paeoniaceae) [10]. PGG exerts robust antioxidant, anti-inflammatory, antiviral, antimicrobial, anti-allergy and anticancer effects [11,12]. In addition, PGG has potential anti-COVID-19 activity by inhibiting the fusion of its key proteins, RBD and angiotensin-converting enzyme-2 (ACE-2) receptors [13]. Previously, we screened the candidate compound PGG from the extract of Qingwen Baidu Decoction using the principal component analysis method for treating an ALI; the compound was classified as *P. suffruticosa* Andr. (Paeoniaceae) by high-pressure liquid chromatography. PGG was further isolated from *P. suffruticosa* Andr. (Paeoniaceae) and its structure was comprehensively characterized. A further spectrum analysis revealed that PGG could be a candidate drug for treating an ALI [14,15]. Therefore, we systematically explored the effects of PGG on an ALI in rats and explored the underlying mechanisms of action.

## 2. Results

### 2.1. PGG Exerts Protective Effects against ALI in Rats

To evaluate the effect of PGG on an ALI, we quantified several important ALI-related indicators, including the total leukocyte count and lung wet/dry (W/D) ratio. As shown in Figure 1A, few cells were detected in the bronchoalveolar lavage fluid (BALF) after an intratracheal saline instillation in the control group, whereas the LPS stimulation resulted in considerable neutrophils and total leukocytes in the BALF. Compared with the LPS group, the PGG (60 and 30 mg/kg)- and dexamethasone (DEX)-treated groups showed a significantly suppressed cell infiltration (*p* < 0.01). In Figure 1B, as the key parameter of edema, the W/D ratio was higher in the LPS-challenged group than in the control group (*p* < 0.01). Interestingly, the PGG (60, 30, and 15 mg/kg)- and DEX-treated groups exhibited a significantly reduced lung edema when compared with the LPS group (*p* < 0.01). 

The oxidative indices, including SOD and MDA, were assessed to measure the antioxidative effects of PGG in LPS-stimulated ALI rats [16,17]. An LPS stimulation decreased the SOD activity, whereas the MDA content was increased. Interestingly, the PGG (60 and 30 mg/kg)- and DEX-treated groups demonstrated an effectively improved SOD activity (Figure 1C) and decreased MDA levels (Figure 1D) (*p* < 0.05). These results suggested that PGG could afford therapeutic effects against an ALI, primarily via its potent antioxidant properties. To assess the effects of PGG on an inflammation, we examined the expression of the essential inflammatory cytokines IL-1β and IL-6 in rats with an LPS-induced ALI. As shown in Figure 1E,F, the levels of inflammatory cytokines IL-1β and IL-6 were evidently higher after the LPS challenge when compared with the control group (*p* < 0.01). However, the treatment in the PGG (60 and 30 mg/kg)- and DEX-treated groups was found to significantly attenuate the production of IL-1β and IL-6 (*p* < 0.05) compared with that observed in the LPS only treated rat.

### 2.2. PGG Alleviates LPS-Induced Pathological Changes in Lung Tissues

To further confirm the therapeutic effect of PGG, lung tissue was harvested from each group and stained with hematoxylin-eosin (H&E). No distinct changes were detected in the lung tissue of normal rats (Figure 2A). We observed significant pathological changes following an LPS stimulation, including an alveolar hemorrhage, swelling of alveolar walls and marked neutrophil infiltration into the alveoli (Figure 2B). However, a treatment with PGG (60 and 30 mg/kg) and DEX significantly alleviated these LPS-induced pathological changes (*p* < 0.01) (Figure 2C–F). 

The lung injury score (LIS) was calculated to examine the histological changes in the PGG-treated ALI tissues (Figure 2G). The total surface score of the slides was based on four categories: alveolar congestion, hemorrhage, neutrophil infiltration and thickness of the alveolar wall. Each category was rated 0–4 as follows: 0 = no injury; 1 = 25% damage to the field; 2 = 50% injury field of the total surface; 3 = 75% of the field with injury; and 4 = diffuse damage [18]. The LIS at 12 h after the LPS perfusion was significantly increased (*p* < 0.01). However, the degree of the lung injury was significantly improved in the PGG (60 and 30 mg/kg)- and DEX-treated groups when compared with that in the LPS group (*p* < 0.01).

### 2.3. Western Blot Analysis 

The protein expression levels of the adenosine monophosphate-activated protein kinase (AMPK), adenosine monophosphate-activated protein kinase phosphorylation (p-AMPK), phosphoinositide 3-kinase (PI3K), protein kinase B (Akt), protein kinase B phosphorylation (p-Akt) and nuclear factor erythroid 2-related factor (Nrf2), six critical proteins were quantified using Western blotting. Compared with the control group, the expression of p-AMPK/AMPK, PI3K, pAkt/Akt and Nrf2 were reduced in the ALI lung tissues. In contrast, the PGG groups (60 and 30 mg/kg) exhibited a markedly higher protein expression than the LPS group (*p* < 0.05). A statistical analysis of the images further confirmed the Western blotting results (Figure 3). 

### 2.4. Effect of PGG on Expression of AMPK, PI3K, Akt, and Nrf2 in Lung Tissues by Immunohistochemical Analysis 

In particular, we analyzed the expression of AMPK, PI3K, Akt and Nrf2 proteins in lung tissues by immunohistochemical staining to elucidate the potential role of PGG in ALI rats. Compared with the control group, the expression of AMPK, PI3K, Akt and Nrf2 were significantly decreased under an LPS stimulation; these activities were significantly increased by the PGG treatment, especially in the PGG (30 and 60 mg/kg) groups (Figure 4). This result was consistent with the data obtained from the Western blotting analysis. 

### 2.5. Immunofluorescence Assay

As shown in Figure 5, the protein expression levels of AMPK, PI3K, Akt and Nrf2 were downregulated in the LPS group, whereas the PGG group (60 and 30 mg/kg) showed the marked upregulation of these expression levels, as determined by the immunofluorescence assay; these findings were in line with those of the Western blotting assay and immunohistochemical analysis. Collectively, these results suggest that PGG may exhibit a therapeutic potential in an ALI via the antioxidant AMPK/PI3K/Akt/Nrf2 pathway.

## 3. Discussion

The recruitment of leukocytes, such as neutrophils and inflammatory responses, are early events in an ALI. An important feature of an ALI is the alveolar inflow of serous fluid, which induces edema [19]. Therefore, the lung W/D ratio, total cells, inflammatory cytokines and histological changes are key indicators of an ALI. Herein, we found that an LPS instillation into the lung caused severe pathological changes, primarily increased the lung permeability and induced edema, as evidenced by an increase in the W/D ratio. Our results revealed that PGG (60 and 30 mg/kg) profoundly inhibited a lung edema and attenuated lung histological changes (*p* < 0.05). The PGG (60 and 30 mg/kg) treatment also reduced the number of total cells and the expression of the essential inflammatory cytokines IL-1β and IL-6 in the BALF (*p* < 0.05). 

An imbalance in the antioxidant system and excessive free radical generation are the main causes of a lipid peroxidation in biofilms and severe cell damage, resulting in an ALI [17]. MDA, an important indicator of oxidative stress, is the final product of polyunsaturated fatty acids and is used to reflect the level of cellular damage caused by the production of ROS metabolites [20]. SOD is a key enzyme that measures the oxygen free radical scavenging ability and exhibits various activities, especially anti-inflammatory and antioxidant effects [21]. In the present study, PGG (60 and 30 mg/kg) alleviated oxidative stress mainly by reducing the production of MDA and elevating the SOD levels (*p* < 0.05). Our results were consistent with the potent antioxidant properties of PGG, a polyphenol substance [10].

It is well-known that Nrf2 can lower the ROS levels and suppresses an oxidative imbalance [22]. Moreover, the PI3K/Akt and AMPK pathways are widely speculated to play key roles in regulating an Nrf2-dependent transcription. Following the induction of oxidative stress, Nrf2 enters the nucleus to activate the antioxidant enzyme SOD and antioxidant gene (HO-1), ultimately reducing the oxidative damage. Based on our findings, PGG exerts a robust antioxidant effect and can afford a protection against an ALI by diminishing the formation of MDA and the depletion of SOD. Therefore, we examined the mechanism of PGG in an ALI using the antioxidative stress-related proteins AMPK, PI3K, Akt and Nrf2.

AMPK, as a sensor of the cellular energy status, regulates the cellular and systemic energy balance [23]. The activation of AMPK has been associated with antioxidant and anti-inflammatory effects in different models. Therefore, protein kinases play diverse roles in mediating cytoprotection against different cellular aspects [24]. A stimulation with specific activators could significantly inhibit the release of pro-inflammatory cytokines, suggesting that the activation of the AMPK protein could inhibit the process of a lung inflammation [25]. This effect may be regulated by transcription factors downstream of Nrf2 [26]. AMPK can activate the PI3K/Akt pathway, and the phosphorylation of Akt further activates the nuclear translocation of Nrf2 [27,28]. Herein, we observed that PGG (60 and 30 mg/kg) upregulated p-AMPK/AMPK expression (*p* < 0.05). The immunohistochemical and immunofluorescence results confirmed those of Western blot analysis.

The PI3K/Akt pathway can regulate apoptosis to facilitate a cell survival and growth [29,30]. Apoptosis plays an important role in the pathogenesis and treatment of ALI. The activation of the PI3K/Akt pathway can significantly decrease the apoptosis of type II pneumocytes and mitigate the BAX protein expression, affording protection against an ALI [31,32]. Based on Western blot results, PI3K and p-Akt/Akt exhibited higher expression levels in the PGG group (60 and 30 mg/kg) than in the model group (*p* < 0.05), revealing that the effect of PGG on an ALI was positively correlated with the activation of the PI3K/Akt signaling pathway.

Nrf2, the core regulatory factor of the cell defense against various stress injuries, is also a key endogenous antioxidant pathway in the body. Following the induction of oxidative stress, Nrf2 rapidly enters the nucleus to initiate SOD activity, reducing the oxidative damage [33,34]. Nrf2 is phosphorylated at serine and threonine residues by kinases, including PI3K, which is beneficial for Nrf2 and its subsequent release translocation [35,36]. AMPK can activate the PI3K/Akt pathway, and the phosphorylation of AKT can further activate the Nrf2 nuclear translocation. In addition, it has been reported that the PI3K/Akt pathway is a putative regulator of the transcription factor Nrf2. The Western blot results also revealed that the expression of the Nrf2 protein was significantly higher in the PGG group (60 and 30 mg/kg) than in the LPS group (*p* < 0.05). The immunohistochemical and immunofluorescence results confirmed the Western blot findings. 

As we known, dexamethasone (DEX) is a commonly used glucocorticoid, which has anti-inflammatory, antiviral and anti-allergic effects. Clinical studies have shown that dexamethasone can save the lives of critically ill patients with COVID-19 [37]. In addition, DEX has been used in ALI/ARDS to improve oxygenation by decreasing the lung collagen and edema formation. However, an improper dosing and medication duration also increase the risk of mortality [6]. Our results showed that both PGG and DEX attenuated an LPS-induced ALI by reducing inflammation, oxidative stress and pulmonary edema. PGG exerts protective effects against an ALI in rats. PGG belongs to the tannin group and exerts robust antioxidant properties. Interestingly, our results showed that PGG can activate the antioxidant pathway to treat an ALI. Our aim was to explore the therapeutic effect of PGG on an ALI and its potential mechanism, with a view to developing innovative drugs for an ALI.

In summary, the candidate compound PGG can activate the AMPK/PI3K/AKT/Nrf2 antioxidant pathway to reduce the proinflammatory cytokines (IL-6 and IL-1β), the MDA formation and SOD consumption, further lowering the alveolar inflammatory cell infiltration and pulmonary edema, and ultimately ameliorating ALI (Figure 6).

## 4. Materials and Methods

### 4.1. Compound 

As described in our previous report, PGG was isolated from *P. suffruticosa* Andr. (Paeoniaceae) [15]. The identification of PGG is detailed in the Appendix A.

### 4.2. Animals

Male Sprague Dawley rats (200–220 g) were obtained from Vital River Laboratories Co., Ltd. (Beijing, China). The animal experiments were approved by the Ethics Committee of Fujian Medical University (No. 2021-J-0547). All the rats were acclimated for 5 days under standard laboratory conditions before the experimentation (22 ± 2 °C temperature, 45–65% relative humidity, 12 h light/dark cycle).

### 4.3. Experimental Design

The rats were randomly divided into six groups: (1) the control group, (2) LPS group, (3) DEX (Tianjin King York Group Co., Ltd., Tianjin, China), (4) PGG (60 mg/kg) + LPS group (HD), (5) PGG (30 mg/kg) + LPS group (MD) and (6) PGG (15 mg/kg) + LPS group (LD). PGG (PGG in 1 mL saline/100 g body weight) was administered intraperitoneally (48, 24, and 1 h) before an LPS administration. DEX (5 mg/kg) was intraperitoneally injected 1 h before the LPS administration (Sigma-Aldrich, St. Louis, MO, USA). The ALI model rats were established by an intratracheal instillation of LPS (5 mg/kg), with normal saline employed in the control group. The PGG dosages were based on our previous study and pre-experimental results [15]. The rats were sacrificed 12 h after the instillation of LPS, and the BALF and lung tissues were collected.

### 4.4. Measurement of Cell Counts in BALF

After the experiment, the right lung was ligated, and the left lung was lavaged with phosphate-buffered saline (PBS; 1 mL/ time, 5 times) for the BALF collection. The recovered BALF volume was 4 mL. The BALF sample was centrifuged at 300× *g*, and the BALF supernatant was stored at −80 °C. The total number of cells was measured using an automatic hematology analyzer after resuspending the cell pellet in PBS.

### 4.5. Lung Wet/Dry (W/D) Ratio

Pulmonary edema was determined by assessing the W/D weight ratio of the lung tissues. After euthanasia, the right upper lung was harvested and immediately weighed as wet weight and then dried to achieve a constant weight to obtain the dry weight. Finally, the W/D ratio was calculated.

### 4.6. Evaluation of Oxidative Stress in BALF 

MDA and SOD are key indices of the lipid oxidation in an ALI [16]. The MDA content and SOD activity were determined using the respective kit instructions (Nanjing Jiancheng Institute of Biotechnology, Nanjing, China). 

### 4.7. Measurement of Cytokine Levels in BALF 

The BALF sample was used to determine the expression of IL-1β and IL-6. The expression of L-1β and IL-6 was determined using the respective kit instructions using ELISA Kits (Shanghai Yuanju Biotechnology Center, Shanghai, China). Assays for each of the target cytokines were conducted in accordance with the instructions provided by the manufacturer, with absorbance in each case being determined at 450 nm.

### 4.8. Histological Examination 

The lung tissues were harvested and fixed in 10% neutral formalin for 12 h following the instillation of LPS. Subsequently, the lung tissues were sliced (5 μm thick) after paraffin embedding and stained with H&E. Finally, the pathological changes in the lungs were observed under a microscope.

### 4.9. Western Blot Analysis

Western blotting was performed on an equal number of protein samples to verify the proteomic data. Briefly, the lung tissue samples, including 30 μg of protein, were isolated by sodium dodecyl sulfate-polyacrylamide gel electrophoresis (SDS-PAGE) (10%) and transferred to PVDF membranes. The membranes were blocked with 5% bovine serum albumin (BSA) for 2 h and incubated overnight with primary antibodies (rabbit anti-PI3Kinase p85 alpha antibody) (PI3Kp85α; Abcam, Cambridge, UK), rabbit anti-AKT pan mAb, AMP-activated protein kinase (AMPKα), Phospho-AMPKα (p-AMPK), Phospho-Akt (p-Akt) and NrF2 (Cell Signaling Technology, Danvers, MA, USA), diluted to 1:1000; β-actin rabbit polyclonal antibody was obtained from DGCS-BIO (Beijing, China; 1:5000). After washing, the membranes were incubated with peroxidase-conjugated goat anti-rabbit IgG (1:5000) for 1 h. The quantification was performed using the BIO-RAD Imaging System software. 

### 4.10. Immunohistochemistry for Detection of PI3K, Akt, Nrf2, and AMPK

The expression of PI3K, Akt, AMPK and NrF2 in the lung tissues was further evaluated by immunohistochemistry. The deparaffinized paraffin sections were rehydrated in graded ethanol, microwaved in sodium citrate buffer and incubated with 3% H_2_O_2_ after cooling. The sections were blocked with 3% BSA (25 °C, 30 min) and then incubated with PI3Kp85α, AkT, AMPKα and Nrf2 overnight at 4 °C. All the sections were washed with PBS and incubated using a secondary antibody (37 °C, 50 min). The samples were then incubated with AEC substrates and counterstained with a hematoxylin staining solution (in the dark). After the nuclear microscopy examination, the film was sealed. Finally, the slides were observed under a microscope.

### 4.11. Immunofluorescence

Immunofluorescence staining was performed to assess the expression of PI3K, Akt, AMPK and Nrf2. The previous steps were performed as those used for immunohistochemistry. The simplified steps are as follows: the lung tissue was deparaffinized, hydrated with alcohol, then incubated with an EDTA antigen retrieval buffer (pH 8.0), washed with PBS, blocked with 3% BSA, incubated with PI3K, AkT, AMPK and Nrf2 overnight at 4 °C, and stained with a suitable fluorescent secondary antibody for 50 min in the dark. Finally, the nuclei were stained with DAPI for 10 min and observed under a fluorescence microscope. The DAPI-stained nuclei were indicated in blue. The positive cells were green, red or pink, depending on the fluorescent label used. 

### 4.12. Data Analysis

All the experimental data are expressed as mean ± standard deviation (SD). Data analyses were performed using SPSS software (SPSS Inc., Chicago, IL, USA) and GraphPad Prism 6.0 Software (GraphPad Software Inc., San Diego, CA, USA). Comparisons between the experimental groups were conducted using a one-way analysis of variance (ANOVA). *p* < 0.05 or *p* < 0.01 was considered statistically significant.

## 5. Conclusions

In conclusion, our findings, for the first time, demonstrated that PGG can effectively inhibit an inflammation and oxidative stress to treat an ALI, which positively correlated with the activation of the AMPK/PI3K/AKT/Nrf2 signaling pathway.

## Figures and Tables

**Figure 1 ijms-23-14423-f001:**
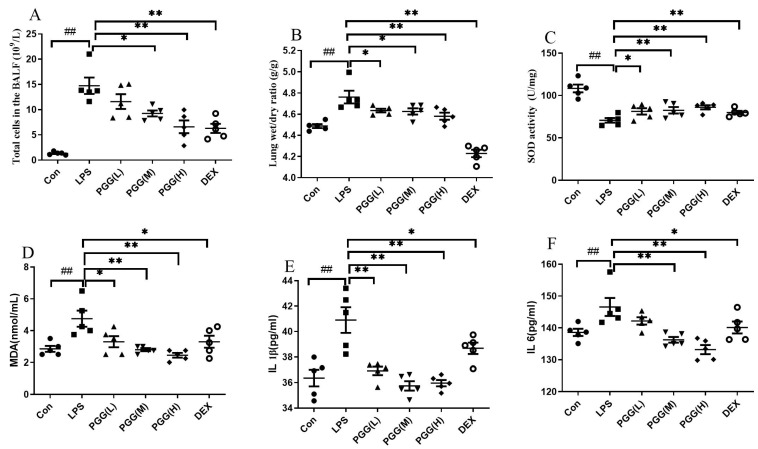
Effects of PGG on anti-inflammatory and antioxidative indicators in ALI rats. DEX and PGG were administered intraperitoneally for three days, and the ALI model was established 1 h after the last administration; animals were sacrificed 12 h later. (**A**) Total leukocyte counts in the BALF; (**B**) lung W/D ratio; (**C**) value of MDA in BALF; (**D**) value of SOD in BALF; (**E**) the IL-1β level in BALF; (**F**) the IL-6 level in BALF. Values are indicated as the mean ± standard deviation (SD). ^##^ *p* < 0.01 vs. Con group; * *p* < 0.05, ** *p* < 0.01 vs. LPS group. ALI, acute lung injury; BALF, bronchoalveolar lavage fluid; Con, control group; DEX, dexamethasone; LPS, lipopolysaccharide; MDA, malondialdehyde; PGG, 1,2,3,4,6-O-pentagalloylglucose; PGG (LD), PGG (15 mg/kg) + LPS group; PGG (MD), PGG (30 mg/kg) + LPS group; PGG (HD), PGG (60 mg/kg) + LPS group; SOD, superoxide dismutase; W/D, wet/dry.

**Figure 2 ijms-23-14423-f002:**
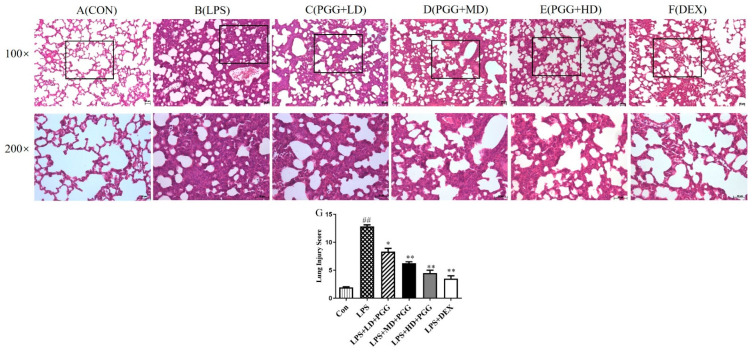
PGG alleviates LPS-induced pathological changes in lung tissue (*n* = 5). Representative lung tissue by hematoxylin-eosin (H&E) staining in each group (**A**–**G**). Lung tissue sections were stained with H&E for histopathologic analysis (magnification 100×, 200×, *n* = 5). Scale bar, 50 μm. (**A**) Control group, (**B**) LPS group, (**C**) PGG (LD) group, (**D**) PGGG (MD) group, (**E**) PGG (HD) group, (**F**) DEX group, (**G**) lung Injury Score. Values are expressed as the mean ± standard deviation (SD). ^##^ *p* < 0.01 vs. Con group; * *p* < 0.05, ** *p* < 0.01 vs. LPS group. Con, control group; DEX, dexamethasone; LPS, lipopolysaccharide; PGG, 1,2,3,4,6-O-pentagalloylglucose; PGG (L), PGG (15 mg/kg) + LPS group; PGG (M), PGG (30 mg/kg) + LPS group; PGG (H), PGG (60 mg/kg) + LPS group.

**Figure 3 ijms-23-14423-f003:**
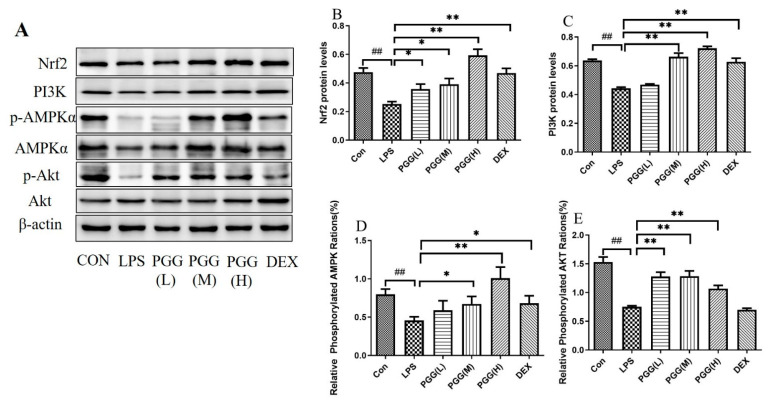
Effect of PGG on expression levels of p-AMPK/AMPK, PI3K, p-Akt/Akt and Nrf2 in lung tissues of ALI rats. (**A**) The protein levels of p-AMPK/AMPK, PI3K, p-Akt/Akt and Nrf2 in lung tissues were determined by Western blotting; (**B**) The expression of Nrf2 in lung tissues; (**C**) The expression of PI3K in lung tissues; (**D**) The expression of p-AMPK/AMPK in lung tissues; (**E**) The expression of p-Akt/Akt in lung tissues. Representative Western blot images of four proteins are altered in the LPS-challenged PGG- and DEX-treated groups compared with the LPS group. Statistical analysis of the protein expressions of p-AMPK/AMPK, PI3K, p-Akt/Akt and Nrf2 was performed. β-actin was used as the reference. Values are expressed as the mean ± standard deviation (SD) (*n* = 3). ^##^ *p* < 0.01 vs. Con group; * *p* < 0.05, ** *p* < 0.01 vs. LPS group. P-Akt, protein kinase B phosphorylation; AKT, protein kinase B; ALI, acute lung injury; p-AMPK, adenosine monophosphate-activated protein kinase phosphorylation; AMPK, adenosine monophosphate-activated protein kinase; Con, control group; DEX, dexamethasone; LPS, lipopolysaccharide; Nrf2, nuclear factor erythroid 2-related factor; PGG, 1,2,3,4,6-O-pentagalloylglucose; PGG (LD), PGG (15 mg/kg) + LPS group; PGG (MD), PGG (30 mg/kg) + LPS group; PGG (HD), PGG (60 mg/kg) + LPS group; PI3K, phosphoinositide 3-kinase.

**Figure 4 ijms-23-14423-f004:**
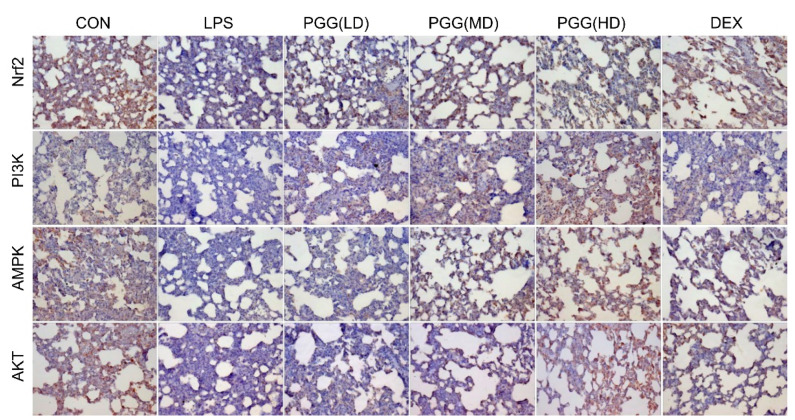
Effect of PGG on PI3K, Akt, Nrf2 and AMPK expressions in the lung tissues by immunohistochemical method. Photomicrographs were obtained at 200×. Akt, protein kinase B; AMPK, adenosine monophosphate-activated protein kinase; CON, control group; DEX, dexamethasone; LPS, LPS-challenged group; Nrf2, nuclear factor erythroid 2-related factor; PGG, 1,2,3,4,6-O-pentagalloylglucose; PGG (LD), PGG (15 mg/kg) + LPS group; PGG (MD), PGG (30 mg/kg) + LPS group; PGG (HD), PGG (60 mg/kg) + LPS group; PI3K, phosphoinositide 3-kinase.

**Figure 5 ijms-23-14423-f005:**
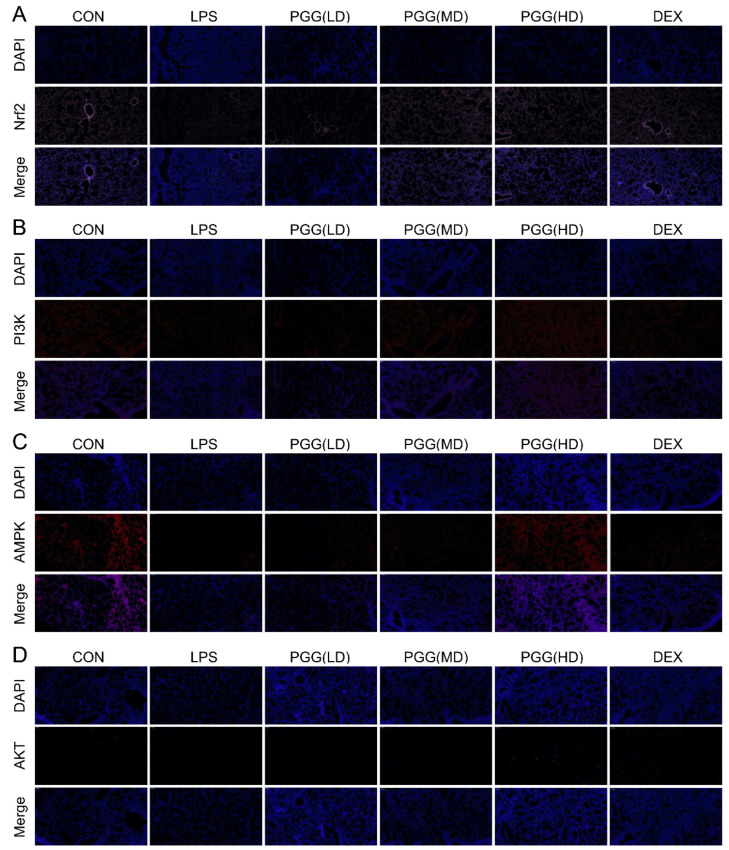
Immunostaining of AMPK, PI3K, Akt and Nrf2 in lung tissue slices. (**A**) Nrf2, (**B**) PI3K, (**C**) AMPK and (**D**) Akt. Photomicrographs were obtained at 100×. Akt, protein kinase B; AMPK, adenosine monophosphate-activated protein kinase; CON, control group; DEX, dexamethasone; LPS, LPS-challenged group; Nrf2, nuclear factor erythroid 2-related factor; PGG, 1,2,3,4,6-O-pentagalloylglucose; PGG (LD), PGG (15 mg/kg) + LPS group; PGG (MD), PGG (30 mg/kg) + LPS group; PGG (HD), PGG (60 mg/kg) + LPS group; PI3K, phosphoinositide 3-kinase.

**Figure 6 ijms-23-14423-f006:**
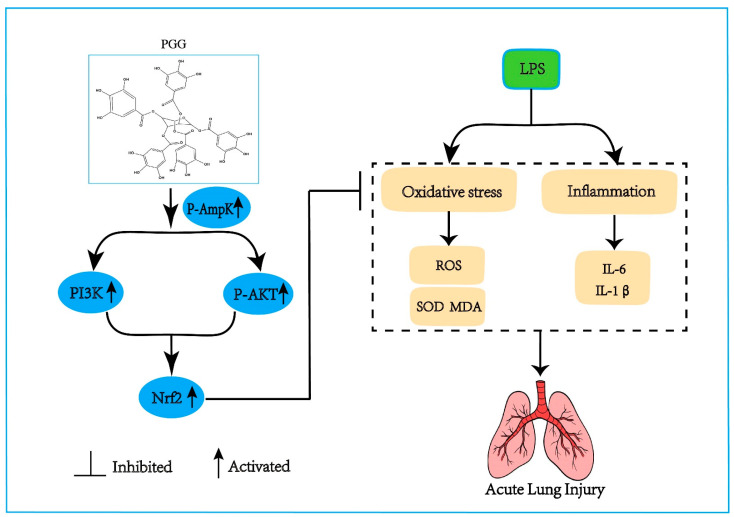
The effect of PGG on ALI and the underlying mechanism. The protective effect of PGG on ALI by inhibiting oxidative stress and inflammation is positively correlated with the activation of the AMPK/PI3K/Akt/Nrf2 pathway. AMPK, adenosine monophosphate-activated protein kinase; p-Akt, protein kinase B phosphorylation; IL-1β, interleukin-1β; IL-6, interleukin-6; LPS, lipopolysaccharide; MDA, malondialdehyde; Nrf2, nuclear factor erythroid 2-related factor; PGG, 1,2,3,4,6-O-pentagalloylglucose; PI3K, phosphoinositide 3-kinase; ROS, reactive oxygen species; SOD, superoxide dismutase.

## Data Availability

The data presented in this study are available upon request from the corresponding author.

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
