# Peer review of "1,2,3,4,6-O-Pentagalloylglucose Protects against Acute Lung Injury by Activating the AMPK/PI3K/Akt/Nrf2 Pathway"

_ijms, 2022, doi:10.3390/ijms232214423_

Round 1

Reviewer 1 Report

The study carried out is very interesting, acute lung injury is a serious lung disease with a high mortality rate, warranting 17 the development of novel therapies. The conclusion demonstrated that PGG can effec-347 tively inhibit inflammation and oxidative stress to treat ALI, which positively correlated 348 with the activation of the AMPK/PI3K/AKT/Nrf2 signaling pathway.  Also, the study is clear and well-structured.

Author Response

Reviewer 1:

The study carried out is very interesting, acute lung injury is a serious lung disease with a high mortality rate, warranting the development of novel therapies. The conclusion demonstrated that PGG can effectively inhibit inflammation and oxidative stress to treat ALI, which positively correlated with the activation of the AMPK/PI3K/AKT/Nrf2 signaling pathway.  Also, the study is clear and well-structured.

Response:

Thank you for your comments.

Reviewer 2 Report

In this manuscript, Zahng et al., have provided a set of data suggesting potential protective effective effect of PGG in LPS induced Acute Lung Injury (ALI) model and underlying action mechanism. Considering ongoing Pandemic, further research in this field alongside identification of new targets and potential therapeutic solutions is pivotal as ALI associated with Covid-19 infection as well as invasive ventilation, remains as one of major causes of death. That being said, there are some major concerns that authors need to address:

General comments:

Author needs to improve the method and material section by satiating number of animals which were used for each group. As a quality of life improvement, they can replace Bar charts with scatter plots to easily represent N for each experiment. Number of animals in each group should be also stated in each figure legend.

Authors need to elaborate on statistical test which were used for each comparison in the figure legends. They need to also expand the statistics section, how they have performed the statistics and not just the software which they have used.

Substantial revision and improvement of discussion is required as authors interpreted increased expression of AMPK, PI3k/AKT and Nrf2 as activation of respective pathways which is not accurate. Increased phosphorylation ratio of AMPK, AKT alongside increase translocation of Nrf2 into nucleus can be considered as activation of each respective pathway. Hence, they need investigate the phosphorylation status of these proteins (please refer to figure 3 comments). Authors needs to also expand discussion further by elaborating potential pros and cons of PGG in comparison to dexamethasone.

 To strengthen data even more, authors should perform Ly6G immunostaining to assess number of neutrophils in the lungs as LPS induced injury mostly mimics neutrophilic inflammatory response. Additionally, they need to also investigate the level of pro- and anti-inflammatory cytokines such as TNF-a, IL-2, IL-6, TGF-b and IL-10 in BALF to address potential anti-inflammatory properties of PGG.  

Figure specific comments:

Figure 2. Authors need to further strengthen presented data by performing stereological morphometry (here is the reference for an extensive review by Christian Mühlfeld and Matthias Ochs, https://doi.org/10.1152/ajplung.00427.2012). Moreover, they need to present low magnification images of lung specimen alongside the high magnification images to visualize larger fields.

Figure 3. Authors need to perform WB for phosphorylated form of AKT as well as AMPK to evaluate activation of AKT and AMPK in all experimental groups. And the result should be presented as phosphorylated/total protein. Ideally, they should also perform WB for Nrf2 using nuclear protein lysate to evaluate shuttling of this transcription factor to nucleus and show nuclear to total protein ratio.

Figure 4. Higher quality images are required as the resolution of images presented in the figure is rather poor. In the line 161 of the manuscript, author have quoted “Compared with the control group, the activities of AMPK, PI3K, AKT, and Nrf2 were significantly decreased under LPS stimulation”, while they only show the expression level which by no means in the case aforementioned proteins correspond to their activity.

Figure 5. contrast and resolution of images are poor. Instead of presenting separate channels and overlay, higher quality overlay images will be sufficient. Additionally, they can present a higher magnification of images. They should also quantify level of fluorescent signal using a software such as Image J to strengthen their data even more. Ideally, Authors should also perform a Ki-67 and TUNEL staining on the sections and assess the proliferation and apoptosis in the experimental groups.

Author Response

Reviewer 2:

Comments and Suggestions for Authors

In this manuscript, Zahng et al., have provided a set of data suggesting potential protective effective effect of PGG in LPS induced Acute Lung Injury (ALI) model and underlying action mechanism. Considering ongoing Pandemic, further research in this field alongside identification of new targets and potential therapeutic solutions is pivotal as ALI associated with Covid-19 infection as well as invasive ventilation, remains as one of major causes of death. That being said, there are some major concerns that authors need to address:

Question 1: General comments:

Author needs to improve the method and material section by satiating number of animals which were used for each group. As a quality of life improvement, they can replace Bar charts with scatter plots to easily represent N for each experiment. Number of animals in each group should be also stated in each figure legend. Authors need to elaborate on statistical test which were used for each comparison in the figure legends. They need to also expand the statistics section, how they have performed the statistics and not just the software which they have used.

Thank you for your comments. We have extensively the manuscript according to your comments and suggestions. The errors you indicated have been revised, further information was supplement in the revision. The detailed corrections are listed and the revised portion are marked up with the "Track Changes". Our reply to questions is below point by point,

Response: According to your comments, we replaced Bar charts with scatter plots to easily represent N for each experiment (Figure.1). We expanded the statistics section including performing the statistics in section 4.12 Data Analysis.

Question2:

Substantial revision and improvement of discussion is required as authors interpreted increased expression of AMPK, PI3k/AKT and Nrf2 as activation of respective pathways which is not accurate. Increased phosphorylation ratio of AMPK, AKT alongside increase translocation of Nrf2 into nucleus can be considered as activation of each respective pathway. Hence, they need investigate the phosphorylation status of these proteins (please refer to figure 3 comments). Authors needs to also expand discussion further by elaborating potential pros and cons of PGG in comparison to dexamethasone.

Response:

Thanks for your comments! Indeed, the interpretation of increased expression of AMPK, PI3k/AKT, and Nrf2 as activation of the respective pathways, as you suggest, requires extensive revision and refinement of the discussion, which is inaccurate. Hence, we investigated the phosphorylation status of AMPK and Akt (Figure 3). Results showed that the PGG groups (60 and 30 mg/kg) exhibited markedly higher protein expression of p-AMPK/AMPK, PI3K, pAkt/Akt, and Nrf2 than the LPS group (P < 0.05).

We also expanded discussion further by elaborating potential pros and cons of PGG in comparison to dexamethasone in the revised manuscript.

Question3:

 To strengthen data even more, authors should perform Ly6G immunostaining to assess number of neutrophils in the lungs as LPS induced injury mostly mimics neutrophilic inflammatory response. Additionally, they need to also investigate the level of pro- and anti-inflammatory cytokines such as TNF-a, IL-2, IL-6, TGF-b and IL-10 in BALF to address potential anti-inflammatory properties of PGG.  

Response:

Thanks for your comments! Ly6G immunostaining is a good method to assess the number of neutrophils in the lung. In the later experiment, we will use the method you suggested to detect neutrophils. Additionally, we investigated the level of pro-inflammatory cytokines including IL-1 β and IL-6 in BALF to address potential anti-inflammatory properties of PGG (Figure 1 E and 1F).  

Question4:

Figure specific comments:

Figure 2. Authors need to further strengthen presented data by performing stereological morphometry (here is the reference for an extensive review by Christian Mühlfeld and Matthias Ochs, https://doi.org/10.1152/ajplung.00427.2012). Moreover, they need to present low magnification images of lung specimen alongside the high magnification images to visualize larger fields.

Response:

Thanks for your comments! According to your suggestion, we have changed Figure 2. We have presented a low magnification image of a lung tissue specimen at 10X and a high magnification image at 20X to show a larger field of view.

Question5:

Figure 3. Authors need to perform WB for phosphorylated form of AKT as well as AMPK to evaluate activation of AKT and AMPK in all experimental groups. And the result should be presented as phosphorylated/total protein. Ideally, they should also perform WB for Nrf2 using nuclear protein lysate to evaluate shuttling of this transcription factor to nucleus and show nuclear to total protein ratio.

Response:

According to your comments, we performed WB on phosphorylated forms of AKT and AMPK to assess the activation of AKT and AMPK in all experimental groups. The results have been expressed in terms of phosphorylation/total protein (Figure 3) in the revised manuscript. Later experiments we will perform WB for Nrf2 using nuclear protein lysate to evaluate shuttling of this transcription factor to nucleus and show nuclear to total protein ratio.

Question6:

Figure 4. Higher quality images are required as the resolution of images presented in the figure is rather poor. In the line 161 of the manuscript, author have quoted “Compared with the control group, the activities of AMPK, PI3K, AKT, and Nrf2 were significantly decreased under LPS stimulation”, while they only show the expression level which by no means in the case aforementioned proteins correspond to their activity.

Response:

Thanks for your comments! According to your suggestion, we have updated the resulting expression in Figure 4. Compared with the control group, the expressions of AMPK, PI3K, Akt and Nrf2 were significantly decreased after LPS stimulation. However, PGG treatment significantly increased the expression of these proteins, especially in the PGG (30 and 60 mg/kg) groups. For the image resolution in in Figure 4, the resolution becomes lower due to the image compression.

Question7:

Figure 5. contrast and resolution of images are poor. Instead of presenting separate channels and overlay, higher quality overlay images will be sufficient. Additionally, they can present a higher magnification of images. They should also quantify level of fluorescent signal using a software such as Image J to strengthen their data even more. Ideally, Authors should also perform a Ki-67 and TUNEL staining on the sections and assess the proliferation and apoptosis in the experimental groups.

Response:

Thanks for your comments! We are so sorry! The contrast and resolution of the images were as sharp as we could try. In Figure 5, we present both the individual channels and the overlay image, and the overlay image is in the "merge" section. According to your suggestion, we will perform Ki-67 and TUNEL staining on the sections of the experimental group in subsequent experiments, and evaluate their proliferation and apoptosis.

Round 2

Reviewer 2 Report

Authors have successfully addressed my major scientific concerns.

Author Response

.